# Effects of Mask Material on Lateral Undercut of Silicon Dry Etching

**DOI:** 10.3390/mi14020306

**Published:** 2023-01-25

**Authors:** Yongkang Zhang, Zhongxuan Hou, Chaowei Si, Guowei Han, Yongmei Zhao, Xiaorui Lu, Jiahui Liu, Jin Ning, Fuhua Yang

**Affiliations:** 1Engineering Research Center for Semiconductor Integrated Technology, Institute of Semiconductors, Chinese Academy of Sciences, Beijing 100083, China; 2Center of Materials Science and Optoelectronics Engineering, University of Chinese Academy of Sciences, Beijing 100049, China; 3School of Electronic, Electrical and Communication Engineering, University of Chinese Academy of Sciences, Beijing 100049, China; 4State Key Laboratory of Transducer Technology, Chinese Academy of Sciences, Beijing 100083, China

**Keywords:** silicon dry etching, undercut, mask materials, Bosch process

## Abstract

The silicon etching process is a core component of production in the semiconductor industry. Undercut is a nonideal effect in silicon dry etching. A reduced undercut is desired when preparing structures that demand a good sidewall morphology, while an enlarged undercut is conducive to the fabrication of microstructure tips. Undercut is related to not only the production parameters but also the mask materials. In this study, five mask materials—Cr, Al, ITO, SiN_x_, and SiO_2_—are chosen to compare the undercut effect caused by the isotropic etching process and the Bosch process. In the Bosch process, the SiN_x_ mask causes the largest undercut, and the SiO_2_ mask causes the smallest undercut. In the isotropic process, the results are reversed. The effect of charges in the mask layer is found to produce this result, and the effect of electrons accumulating during the process is found to be negligible. The undercut effect can be enhanced or suppressed by selecting appropriate mask materials, which is helpful in the MEMS process. Finally, using an Al mask, a tapered silicon tip with a top diameter of 119.3 nm is fabricated using the isotropic etching process.

## 1. Introduction

Silicon etching is a vital process in the fabrication of many microstructures and microelectromechanical systems, such as gyros [1], accelerometers [2], resonators [3], field emission arrays [4], probes [5], etc. Compared with chemical-based wet etching methods, dry etching methods—mainly plasma etching—have the advantages of low damage and good shape controllability [6]. Thus, plasma etching has become the main technology for silicon etching. Silicon dry etching can be divided into two types: isotropic and anisotropic. Anisotropic etching, or deep reactive ion etching (DRIE), is the preferred process for high-aspect-ratio MEMS structures such as grooves, vias, or pillars. The DRIE process achieves good anisotropy through a combination of etching and passivation mechanisms. Fluorine radicals are the main component in the chemical process of the etching reaction. The decomposition of SF_6_ and other fluorine-based gases produces fluorine free radicals, and then the radicals absorb and remove the silicon. C_4_F_8_ and other fluorocarbon-based gases are mainly used in the passivation process by forming a passivation layer composed of Teflon on the silicon surface. Passivation layers such as Teflon can restrain free fluorine radicals from etching the sidewall to decrease the lateral etching rate and improve the anisotropy of the reaction. The isotropic process only includes the etching process in DRIE, allowing it to achieve a higher etching rate and severe lateral etching. It is generally used to etch low-depth trenches (less than 5 µm), release sacrificial layers [7], and form tips [8].

Since the 1980s, the mechanisms [9], components [10], and applications [11] of silicon plasma etching have received extensive research. Most studies focus on the control of the sidewall profile. Undercut is a critical nonideal effect that causes a higher lateral etching rate of the sidewalls beneath masks, which reduces the flatness and uniformity of the sidewall. In many applications, undercut leads to complications in subsequent processing and degraded device performance. In other special applications, such as the preparation of silicon tips, the undercut effect is needed to achieve a small size in the conical pointed structure. Methods to modulate the undercut effect in the silicon etching process have been proposed. The components are a major factor affecting the undercut effect. It has been proven that the ratio of SF_6_ to C_4_F_8_ gas in the mixture etching process [12] and the ratio of SF_6_ to O_2_ gas flows in the cryogenic process [13] can affect the lateral erosion of the undercut. In the Bosch process, the shortened process times of the etching step and the passivation step lead to smaller scallops and less undercut. In addition, the increased chamber pressure leads to a larger undercut [14]. Mask material also contributes to the undercut effect when using the same processes. For example, Al and Cr cause a decreased undercut compared to their oxide [15], and a SU8/SiO_2_ double-layer mask achieves less undercut than a SU8 single-layer mask [13]. The problem is that the mechanism of how mask materials affect undercut is not clear.

In this work, several commonly used mask materials in MEMS fabrication, i.e., Cr [16], Al [15,17], ITO [18], SiN_x_ [19], and SiO_2_ [10,20], are chosen to compare the undercuts caused by the isotropic process and the Bosch process, i.e., the most popular DRIE processes. Undercut characterization is based on optical microscopy (OM) and scanning electron microscopy (SEM). By analyzing the experimental results, we find that the charged mask plays a leading role in the passivation step, and the undercut can be adjusted by selecting appropriate mask materials. Finally, by using an Al mask to maximize the undercut, a tapered silicon tip with a top diameter of 119.3 nm is fabricated using isotropic etching. This work contributes to our development of the techniques applied in the preparation of MEMS devices.

## 2. Experiments

Four-inch, p-type<100>, single-side-polished monocrystalline silicon wafers with a resistivity of 1–10 Ω⋅cm were prepared as substrates. Cr, ITO, and Al with a thickness of 50 nm were coated on the wafer using magnetic sputtering and then patterned with a lift-off process using a 2 µm thick L300 negative photoresist as a sacrificial layer. SiO_2_ and SiN_x_ films were deposited on the wafer with plasma-enhanced chemical vapor deposition (PECVD) with a thickness of 1120 nm, and then a 2 µm thick Az6130 positive photoresist was spin-coated and patterned using lithography. An etching process using Sentech ICP-RIE SI 500 in a gas flow of CHF_3_ was performed to transfer the patterns from the photoresist to the silicide film. Once the etching was completed, the remaining photoresist was stripped. Parameters for the SiO_2_ and SiN_x_ etching processes are shown in Table 1.

The mask pattern was a parallel square array, as shown in Figure 1. Both the side length and spacing of the square patterns were 20 µm. Measuring the side length of a square produces less error than for a circle, triangle, hexagon, and other geometric shapes. In addition, the periodic arrangement of the square patterns improves the uniformity of etching.

Figure 2 shows a schematic of the undercut effect assessment by ΔL, i.e., the lateral etching length under the mask. The side length of the patterned square mask, marked as $L_{1}$, was measured before silicon etching. After silicon etching and mask removal, the side length of the square on the upper surface of the remaining silicon pillar was measured and marked as $L_{2}$. Then, ΔL could be calculated as:(1)$$\Delta L=\frac{L_{1}-L_{2}}{2}$$

A LEXT OLS4000 microscope was used to measure $L_{1}$ and $L_{2}$. Masks were stripped using a wet etching process: Cr, ITO, and Al were stripped with boiled sulfuric acid/peroxide mi (SPM); SiN_x_ was stripped with a hot H_3_PO_4_ solution; and SiO_2_ was stripped with buffered hydrofluoric acid (BHF).

The Bosch process is usually applied to DRIE silicon etching. As a time-multiplexed process alternating etching and passivation steps, the Bosch process achieves a higher aspect ratio than mixture processes and still maintains a high etching rate [21]. In this work, a two-step Bosch process was adopted. The process cycle began with the passivation step, and C_4_F_8_ gas flow was used to form a fluorocarbon polymer layer on all exposed surfaces. The etching step was subsequently carried out, and SF_6_ gas flow was used to remove Si isotropically using the reaction between Si and the produced fluorine free radicals. The wafer-chuck bias source of the ICP plasma etcher was usually opened up in the etching stage to increase the anisotropy of etching. The RF bias voltage produced by the wafer-chuck bias source provided downward vertical velocity for ions and free radicals and enhanced their physical sputtering to the bottom. In the passivation stage, the wafer-chuck bias source should be closed so that passivation layer films can be deposited uniformly. SF_6_ has no chemical reaction with the mask materials. C_4_F_8_ can etch SiO_2_ and SiN_x_ via chemical reactions. In the etching process of masks, fluorocarbon polymers were formed and subsequently removed with physical bombardment. Because of the low RF bias, the etching rate of silicide masks is much lower than the etching rate of silicon. Therefore, all masks can achieve high etching selectivity.

The Bosch process was carried out using an Oxford PlasmaPro 100 ICP plasma etcher. To make it easier to measure the etching data and to show the effect of mask materials on undercut, the process with the maximum lateral etching rate was selected. Throughout the whole process, the chamber temperature was maintained at 180 °C, and the pressure was maintained at 20 mTorr. The passivation step was carried out at an ICP source power of 1500 W and a mixed gas flow of 200 sccm/10 sccm C_4_F_8_/SF_6_. The etching step was carried out at an ICP source power of 1800 W, LF bias power of 4 W, and mixed gas flow of 200 sccm/10 sccm SF_6_/C_4_F_8_. SF_6_ and C_4_F_8_ need to be used alternately, and a fast switching speed is required. The use of mixed gas instead of pure gas can improve the switching speed and ensure process stability and repeatability. Compared to the flow of the main reactant, the gas flow of 10 sccm was very small and had no significant effect on the experimental results. There were 40 cycles in the whole process, and each cycle was 7 s in duration, including a passivation step of 2 s and an etching step of 5 s. The detailed parameters of the Bosch process are shown in Table 2. After the mask removal process, the height of the remaining silicon pillars was measured using a Bruker step profiler, and the etching rate of the Bosch process was calculated.

The Oxford PlasmaPro 100 ICP plasma etcher was also used for the isotropic etching process, which was carried out at a chamber temperature of 180 °C, a pressure of 45 mTorr, an ICP source power of 1750 W, a pure SF_6_ gas flow of 300 sccm, and a duration of 20 s. A higher SF_6_ flow and pressure than in Bosch’s etching step were selected to increase the lateral etching rates. The detailed parameters for the isotropic etching process are shown in Table 3. After the mask removal process, the height of the remaining silicon pillars was measured using the same Bruker step profiler. Then, the etching rate of the isotropic process was calculated.

High lateral erosion of the isotropic etching process is conducive to the fabrication of microstructure tips. A parallel circular array was selected as the mask pattern. Both the diameter and spacing of each circle were 50 µm, as shown in Figure 3.

A 50 nm thick Al mask was coated on the substrate using magnetic sputtering and patterned with a lift-off process. Then, the silicon tip structure was etched with an isotropic etching process using the parameters in Table 3, except the etching time was increased from 20 s to 150 s. After etching, the Al mask was stripped with boiled SPM. Finally, the diameter of the tip was measured with an FEI Helios G4 CX double-beam scanning electron microscope. 

## 3. Results

The parameter $L_{1}$ of different mask materials was measured with a LEXT OLS4000 microscope. Although all kinds of materials were patterned with the same lithography mask, the actual pattern sizes are slightly different because of different pattern processes. In the lift-off process of Cr, ITO, and Al, the negative photoresist should form a toppling structure with a narrow top and a wide bottom after exposure and development, so the materials out of the pattern area can be completely stripped. The process creates larger-sized material mask patterns than the lithographic mask patterns. In contrast, silicide masks are formed by a positive photography-etching process. The pattern process of positive photoresist creates smaller-sized photoresist patterns than the graphic size of the lithographic mask. The measurement results were in agreement with this theoretical expectation. The square sizes at different positions in the pattern array had good consistency.

The average etching depth of the Bosch process is approximately 30 µm, and the average vertical etching rate is 0.75 µm/cycle. Figure 4a shows an SEM image of a silicon pillar sidewall etched with the SiN_x_ mask. After the mask was removed, the value L_2-Bosch_ was measured to calculate ∆L_Bosch_ and the relative deviation. Figure 4b shows an OM image of the sample using the SiN_x_ mask after the Bosch process.

The experimental data from the Bosch process are shown in Table 4. In the Bosch process, the mask material shows a more considerable impact on the undercut effect. The SiN_x_ and ITO masks cause the largest undercut of 0.65 µm, while the SiO_2_ mask has the smallest ∆L_Bosch_ of 0.47 µm. 

The average etching depth of the isotropic process is 4.6 µm, and the average vertical etching rate is 0.23 µm/s. The SiN_x_ mask layer is etched 0.34 µm and the SiO_2_ mask layer is etched 0.07 um, and the etching rates are 17 nm/s and 3.5 nm/s, respectively. Metal mask layers and the ITO mask layer are etched only 1—2 nm. After mask removal, the value $L_{2}$ of the isotropic process is measured and marked as L_2-isotropic_, and then the value ∆L_isotropic_ and relative deviation of different mask materials are calculated. Figure 5 shows the optical microscope (OM) image of the sample using the SiO_2_ mask.

The experimental data from the isotropic etching process are shown in Table 5, revealing that the mask material did not seem to impact the undercut effect. The average undercut of isotropic etching is approximately 3 µm, and the lateral etching rate is approximately 0.15 µm/s. ITO and SiN_x_ cause slightly less undercut than other mask materials. This indicates that the effect of the mask material on the undercut is opposite in the isotropic process and the Bosch process. In the Bosch process, the SiO_2_ mask causes the smallest undercut, while the SiN_x_ and ITO masks cause the largest undercut.

Figure 6 shows an SEM image of the silicon tip fabricated by the isotropic process in this work. The diameter of the tip structure is 119.3 nm. The process can be used for the fabrication of biomedical microneedles, AFM, and other structures because the feature size of silicon tips can be modulated by mask patterns and parameters. 

## 4. Discussion

To explain the mechanism of how mask materials affect undercut, the reaction process of silicon etching should be introduced. Free fluorine radicals produced by SF_6_ gas play a major role in the etching mechanism. The radicals are adsorbed at the surface, and silicon is etched isotropically by the following chemical reaction [22]:(2)$$Si+4F=SiF_{4}$$

The plasma produced by SF_6_ gas also contains heavy ions such as SF_5_^+^. These positive ions are accelerated vertically downward by the RF bias electric field, which causes physical sputtering on the bottom and increases the anisotropy of the etching. In the passivation mechanism, CF_x_ radicals are produced by C_4_F_8_ gas. Then, these ions are adsorbed at the surface, depositing a uniformly fluorocarbon passivation layer to prevent etching. Due to the anisotropy of the etching, the etching rate of the passivation layer on the bottom is higher than that on the sidewall.

Under ideal conditions, the fluorine radicals in the etching process can completely remove the bottom passivation layer while the sidewall passivation layer remains. In the actual process, an imbalance between etching and passivation causes undesirable effects. When undercut occurs, lateral erosion is larger at the top sidewall. This indicates that some mechanism, such as enhanced etching or weakened passivation, causes the etching rate to exceed the passivation layer deposition rate in this area.

The charged mask can affect lateral erosion by deflecting SF_5_^+^ ions. The positive charges in the mask film repel heavy SF_5_^+^ ions and reduce the etching of the passivation layer at the top sidewall. In contrast, the negative charges in the mask film improve the etching rate. Although the plasma still contains F^−^ ions, the negative ion is lighter and has less effect on physical sputtering to the passivation layer.

It has been proven that the deposition of a carbon–fluorine passivation layer is an ion-enhanced process [23]. CF_x_^+^ ions play a major role in the deposition of the passivation layer. The negative charge in the mask film can attract positive ions, which increases the deposition rate of the passivation layer at the top sidewall, while the positive charge reduces the thickness of the passivation layer. Compared with the etching mechanism, the deposition of the passivation layer is more greatly affected by ions. Therefore, undercut is mainly controlled by the passivation step during the DRIE etching process.

The proposed explanation can explain the results of the experiment well. PECVD SiN_x_ usually contains a positive charge of approximately 10^12^ cm^−2^ [24], while PECVD SiO_2_ usually contains a negative charge of approximately 10^12^ cm^−2^ [25]. Some electrons will accumulate on the upper surface of the mask layer during the process [15]. Although the mask is negatively charged by electrons, the number of electrons is very small. Therefore, the effect of electron accumulation in the reaction is negligible. In the isotropic etching process, the physical bombardment of heavy SF_5_^+^ ions mainly affects the lateral erosion of the top sidewall, so the undercut of the SiO_2_ mask is slightly worse than that of the SiN_x_ mask, as shown in Figure 7a,b. In the Bosch process, the passivation step plays a major role, and SiN_x_ causes a more severe undercut effect than SiO_2_, as shown in Figure 7c–f. As a special conductive oxide, the influence mechanism of ITO on undercut is more complex. Figure 8 shows the correlation of undercut with mask charging. 

## 5. Conclusions

In this work, the undercut effect caused by mask materials in silicon dry etching was studied. Cr, Al, ITO, SiN_x_, and SiO_2_ were chosen to compare the undercutting caused by the isotropic etching process and Bosch process. An optical microscope was used to characterize the lateral erosion at the top sidewall. The experimental results show that the mask material has a slight impact on the undercut effect in the isotropic etching process but shows a more considerable impact in the Bosch process. Under the Bosch process, the SiN_x_ mask has the largest undercut, followed by the metal mask, and the SiO_2_ mask has the least undercut. In the case of isotropic etching, the results are reversed. The effect of charges in the mask layer is responsible for this result, while the effect of electrons accumulating during the process is negligible. In different processes, the undercut effect can be enhanced or suppressed by selecting appropriate mask materials. When using an Al mask to achieve the largest undercut, a tapered silicon tip with a top diameter of 119.3 nm was fabricated with isotropic etching. The process can be used for the fabrication of biomedical microneedles, AFM, and other structures. This work contributes to our development of the techniques applied in the fabrication of MEMS devices.

## Figures and Tables

**Figure 1 micromachines-14-00306-f001:**
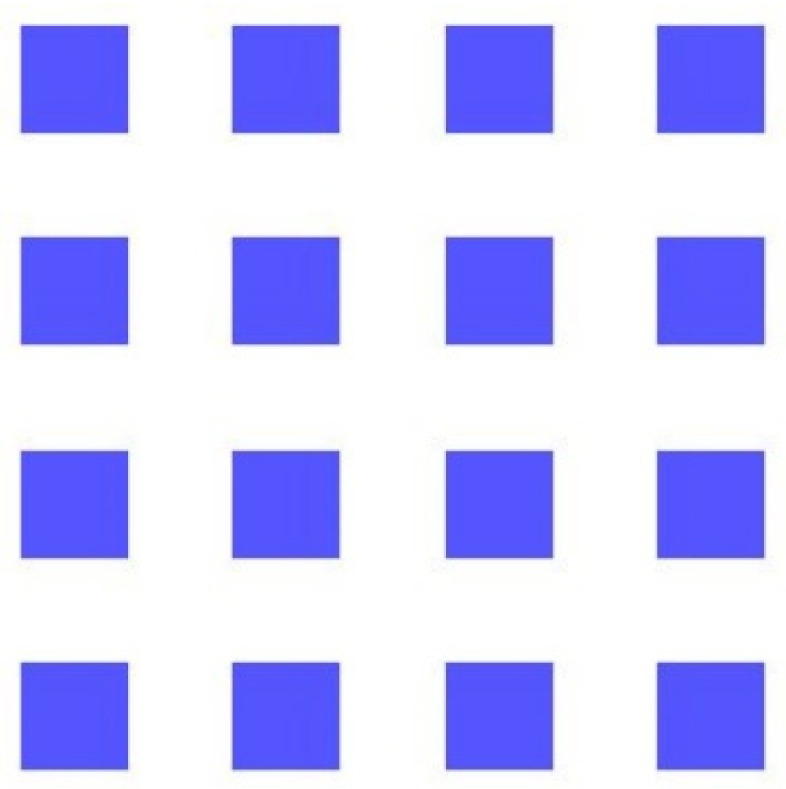
Mask patterns of the 20 µm square array.

**Figure 2 micromachines-14-00306-f002:**
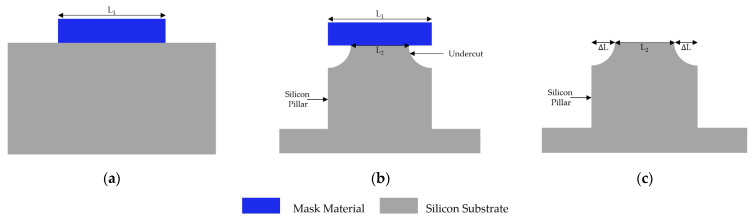
Assessment of undercut. (**a**) *L*_1_ is measured before etching; (**b**) side view of the silicon pillar with the remaining mask; (**c**) *L*_2_ is measured after the mask is removed. Then, Δ*L* is calculated.

**Figure 3 micromachines-14-00306-f003:**
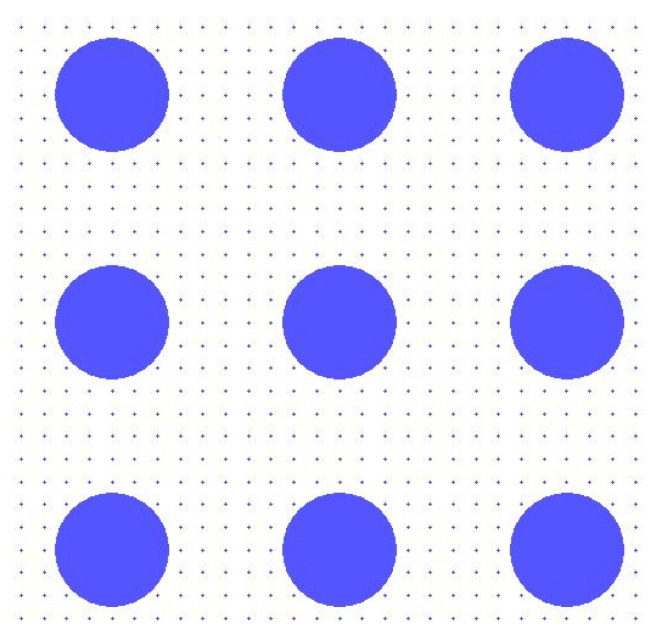
Mask patterns of 50 µm parallel circle array for silicon tip fabrication.

**Figure 4 micromachines-14-00306-f004:**
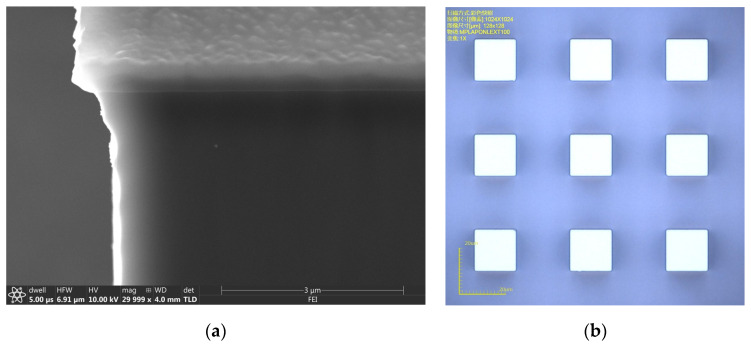
Sample etched with the SiN_x_ mask. (**a**) Sidewall morphology of a silicon pillar with the mask remaining. (**b**) OM image of the sample after etching with the mask removed.

**Figure 5 micromachines-14-00306-f005:**
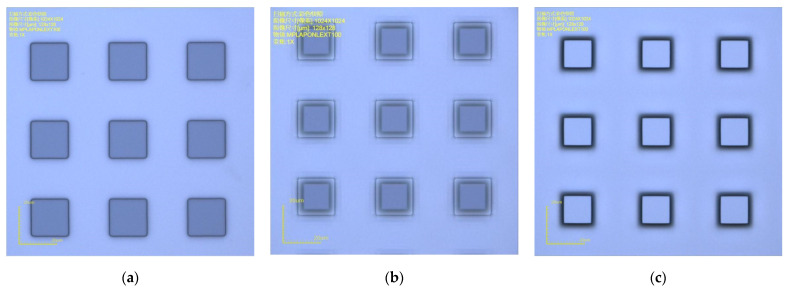
OM images of the sample using the SiO_2_ mask. (**a**) Before etching. (**b**) After etching with the remaining mask. (**c**) After etching with the mask removed.

**Figure 6 micromachines-14-00306-f006:**
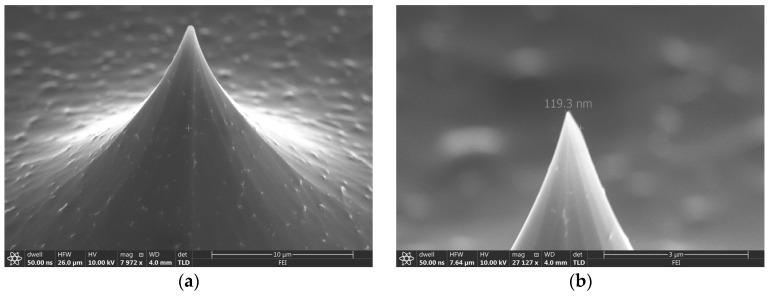
SEM image of a fabricated silicon tip. (**a**) Entire view of the tip. (**b**) Enlarged view of the top position.

**Figure 7 micromachines-14-00306-f007:**
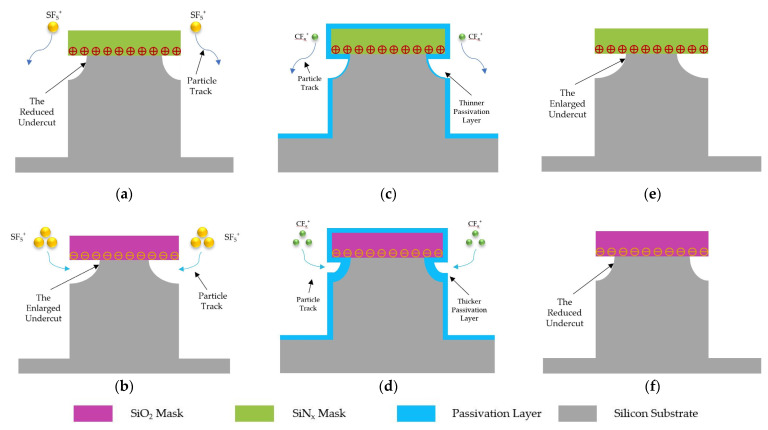
Schematic diagram of a charged mask acting on ions. (**a**) SF_5_^+^ ions are repelled by positive charges. This lowers lateral etching rates and reduces undercut. (**b**) SF_5_^+^ ions are attracted by negative charges. This raises lateral etching rates and an enlarges undercut. (**c**) CF_x_^+^ ions are repelled by positive charges. Then, a thinner passivation layer is formed at the top sidewall. (**d**) CF_x_^+^ ions are attracted by negative charges. Then, a thicker passivation layer is formed at the top sidewall. (**e**) The thinner passivation layer in (**c**) causes an enlarged undercut after the etching process. (**f**) The thicker passivation layer in (**d**) causes a reduced undercut after the etching process.

**Figure 8 micromachines-14-00306-f008:**
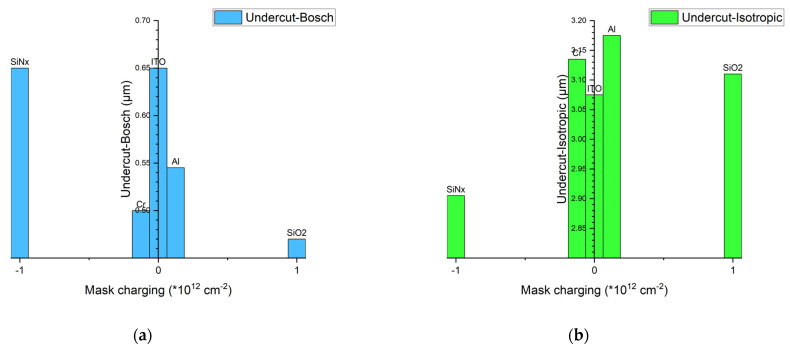
The correlation of undercut with mask charging. (**a**) The Bosch process. (**b**) The isotropic process.

**Table 1 micromachines-14-00306-t001:** Parameters for the silicide mask etching process.

Mask Material	ICP Power	RF Power	CHF_3_ Flow	Pressure	Temperature	Time
Unit	[W]	[W]	[sccm]	[mTorr]	[°C]	[s]
SiO_2_	400	50	60	7.5	20	290
SiN_x_	500	100	60	7.5	20	300

**Table 2 micromachines-14-00306-t002:** Parameters for the Bosch process in this work.

Setting	Unit	Passivation	Etching
Temperature	[°C]	180	180
Pressure	[mTorr]	20	20
SF_6_ tlow	[sccm]	10	200
C_4_F_8_ tlow	[sccm]	200	10
ICP power	[W]	1500	1800
LF power	[W]	0	4
Time	[s]	2	5
Total time	[s]	280	

**Table 3 micromachines-14-00306-t003:** Parameters for the isotropic etching process in this work.

Setting	Temperature	Pressure	SF_6_ Flow	ICP Power	LF Power	Time
Unit	[°C]	[mTorr]	[sccm]	[W]	[W]	[s]
Parameter	180	45	300	1750	0	20

**Table 4 micromachines-14-00306-t004:** Experimental data from the Bosch process. Unit: µm.

Mask Materials	*L* _1_	L_2-Bosch_	ΔL_Bosch_	Relative Deviation
Cr	20.23	19.23	0.500	−11.2%
ITO	21.90	19.60	0.650	15.5%
Al	20.35	19.44	0.545	−3.20%
SiO_2_	19.21	18.27	0.470	−16.5%
SiN_x_	18.90	17.60	0.650	15.5%
Average	20.12	18.83	0.56	32.1% ^1^

^1^ The relative range of ∆L_Bosch_.

**Table 5 micromachines-14-00306-t005:** Experimental data from the isotropic etching process. Unit: µm.

Mask Materials	*L* _1_	L_2-Isotropic_	ΔL_Isotropic_	Relative Deviation
Cr	20.53	14.26	3.135	1.79%
ITO	20.77	14.62	3.075	−0.16%
Al	20.61	14.26	3.175	3.08%
SiO_2_	19.18	12.96	3.110	0.97%
SiN_x_	18.67	12.86	2.905	−5.68%
Average	19.95	13.79	3.080	8.77% ^1^

^1^ The relative range of ∆L_isotropic_.

## Data Availability

Not applicable.

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
