# Peer review of "Effects of Mask Material on Lateral Undercut of Silicon Dry Etching"

_micromachines, 2023, doi:10.3390/mi14020306_

Round 1

Reviewer 1 Report

The results shown in the paper are interesting. It is interesting to see that the lateral undercut depends also on the masking materials. However, a lot of discussion is needed. Here, some comments to improve the manuscript:

1. What is the etching rate of the making layers?

2. Which chemical reactions could one expect between the etchant and the masking materials? Do the reaction products interfere with the etching of silicon?

3. Why do you use the same gases for etching and passivating? It is clear that their proportion is different in one case and in the other, but, what is the reason to use the same two gases?

4. The lateral etching rate is large even in the best case of anisotropic etching. What is the reason?

5. Could it be that the masking materials get charged during etching? You should consider not just the trapped charges  in the starting masking materials.

6. Modify the conclusions and abstract  according to all the changes in the manuscript.

Author Response

Dear Reviewer:

Thank you for your comments concerning our manuscript entitled ‘Effects of mask material on lateral undercut of silicon dry etching (ID: micromachines-2155981). Those comments are all valuable and very helpful for revising and improving our paper, as well as the important guiding significance to our research. We have studied comments carefully and have made correction which we hope meet with approval. All the revisions to the manuscript had been marked up using the “Track Changes” function. The main corrections in the paper and the responds to the reviewer’s comments areas flowing:

Point 1: What is the etching rate of the making layers?

Response 1: In the isotropic process, the SiNx mask layer is etched 0.34um and the SiO2 mask layer is etched 0.07um. The etching rate of SiNx is 17nm/s and the etching rate of SiO2 is 3.5nm/s. Cr, Al, and ITO are only etched approximately 1-2 nm, their etching rates can be negligible. The description has been added on lines 189 to 191.

Point 2: Which chemical reactions could one expect between the etchant and the masking materials? Do the reaction products interfere with the etching of silicon?

Response 2: SF6 has no chemical reaction with all mask materials.  C4F8 can etch SiO2 and SiNx by chemical reactions. In the etching process of silicide masks, fluorocarbon polymers are formed and subsequently removed by physical bombardment. Because of the low RF bias, the etching rate of mask is much lower than the etching rate of silicon. The reaction products have little effect on the etching of silicon. The selected mask materials are commonly used in silicon etching and have high etch selectivity ratio. The selectivity ratio of SiNx is 17.5 and the selectivity ratio of SiO2 is 66. The selectivity ratio of Al, Cr and ITO can reach more than 5000. The description has been added on lines 119 to 123.

Point 3: Why do you use the same gases for etching and passivating? It is clear that their proportion is different in one case and in the other, but, what is the reason to use the same two gases?

Response 3: In the Bosch process, SF6 is used for the etching reaction and C4F8 is used for the passivation reaction. The two gases need to be used alternately, and a fast switching speed is required. The use of mixed gas instead of pure gas can improve switching speed and ensure process stability and repeatability. Compared to the flow of the main reactant, the gas flow of 10sccm is very small and has no significant effect on experimental results. The description has been added on lines 131 to 135.

Point 4: The lateral etching rate is large even in the best case of anisotropic etching. What is the reason?

Response 4: Our research focus on the effect of mask materials on undercut. To make it easier to measure the lateral etching data and to show the effect of mask materials on undercut, the process with the maximum lateral etching rate is selected. The description has been added on lines 125 to 126.

Point 5: Could it be that the masking materials get charged during etching? You should consider not just the trapped charges in the starting masking materials.

Response 5: Some electrons will accumulate on the upper surface of the mask layer during the process. Although the mask is negatively charged by electrons, the number of electrons is very small. Therefore, the effect of electron accumulation in the reaction can be negligible. The description has been added on lines 245 to 248.

Point 6: Modify the conclusions and abstract according to all the changes in the manuscript.

Response 6: The conclusions and abstract have been modified.

We tried our best to improve the manuscript and made some changes in the manuscript. These changes will not influence the content and framework of the paper. And here we did not list the changes but marked in red in revised paper.

We appreciate for your warm work earnestly, and hope that the correction will meet with approval.

Once again, thank you very much for your comments and suggestions.

Yours sincerely,

Yongkang Zhang

Name: Fuhua Yang, Jin Ning

E-mail: fhyang@semi.ac.cn (F.Y.)

ningjin@semi.ac.cn (J.N.)

Reviewer 2 Report

This manuscript reports the effect of mask material on the undercut of silicon dry etching. It was shown that SiO2 mask causes minimum undercut among the investigated mask materials in the Bosch process. The results are reversed in isotropic etch. The authors explain the result with the positive or negative charging of the mask material. The manuscript is overall well written. Following are my suggestions for further improvement.

1.      The authors can add more description on how the plasma parameters for the Bosch process are optimized in this study.

2.      The authors believe that SiNx surface can be positively charged based on ref. 24, where the measurements are performed outside a plasma. Does the SiNx surface remain positively charged inside the plasma? Typically, negative charges will accumulate on surfaces when a plasma turns on.

3.      It would be helpful to include a plot of undercut versus mask charging for all the mask materials used in this study to visualize their correlation.

Author Response

Dear Reviewer:

Thank you for your comments concerning our manuscript entitled ‘Effects of mask material on lateral undercut of silicon dry etching (ID: micromachines-2155981)‘. Those comments are all valuable and very helpful for revising and improving our paper, as well as the important guiding significance to our research. We have studied comments carefully and have made correction which we hope meet with approval. All the revisions to the manuscript had been marked up using the “Track Changes” function. The main corrections in the paper and the responds to the reviewer’s comments areas flowing:

Point 1: The authors can add more description on how the plasma parameters for the Bosch process are optimized in this study.

Response 1: The description has been added on lines 59 to 62. According to the new Ref. 14, the shortened etching and passivation times lead to small undercut. And the high chamber pressure leads to the larger lateral etching.

Point 2: The authors believe that SiNx surface can be positively charged based on ref. 24, where the measurements are performed outside a plasma. Does the SiNx surface remain positively charged inside the plasma? Typically, negative charges will accumulate on surfaces when a plasma turns on.

Response 2: Some electrons will accumulate on the upper surface of the mask layer during the process. Although the mask is negatively charged by electrons, the number of electrons is very small. Therefore, the effect of electron accumulation in the reaction can be negligible. The description has been added on lines 245 to 248.

Point 3: It would be helpful to include a plot of undercut versus mask charging for all the mask materials used in this study to visualize their correlation.

Response 3: The plot has been added on line 265.

We tried our best to improve the manuscript and made some changes in the manuscript. These changes will not influence the content and framework of the paper. And here we did not list the changes but marked in red in revised paper.

We appreciate for your warm work earnestly, and hope that the correction will meet with approval.

Once again, thank you very much for your comments and suggestions.

Yours sincerely,

Yongkang Zhang

Name: Fuhua Yang, Jin Ning

E-mail: fhyang@semi.ac.cn (F.Y.)

ningjin@semi.ac.cn (J.N.)

Round 2

Reviewer 1 Report

Thanks for considering the comments. The manuscript is not acceptable for publication. The open questions were replied.

Just for the future: The materials can be charged not just by electrons, but by ions and functional groups.